# Social network determinants of alcohol and tobacco use: A qualitative study among out of school youth in South Africa

Rachana Desai[1,2]*, Robert A. C. Ruiter[3], Ansuyah Magan[4], Priscilla S. Reddy[1], Liesbeth A. G. Mercken[5]

1 Human & Social Capabilities Division, Human Sciences Research Council, South Africa, 2 Centre of Excellence in Human Development, University of the Witwatersrand, Johannesburg, South Africa, 3 Department of Work & Social Psychology, Maastricht University, Maastricht, The Netherlands, 4 SAMRC/ Wits Developmental Pathways for Health Research Unit, Faculty of Health Sciences, University of the Witwatersrand, Johannesburg, South Africa, 5 Department of Health Promotion, CAPHRI School for Public Health and Primary Care, Maastricht University, Maastricht, The Netherlands

☯ These authors contributed equally to this work.
* rachana.desai@wits.ac.za

**Data Availability Statement:** All relevant data are within the paper and its Supporting Information files.

## Abstract

An important determinant of alcohol and tobacco use is the adolescent's social network, which has not been explored among out of school youth (OSY). OSY are adolescents not currently enrolled in school and have not completed their schooling. This study aims to qualitatively understand how OSY's social networks support or constrain alcohol and tobacco use. Respondent-driven sampling was used to select 41 OSY (aged 13–20 years) for individual in-depth interviews in a South African urban area. The data were analysed using content analysis. Smoking and drinking friends, family close in age to OSY that drank and smoked, and lack of parental support were associated with alcohol and tobacco use among OSY. Household norms, romantic partners and non-smoking or non-drinking friends were suggested to mitigate alcohol and tobacco use. Understanding how the social network of OSY plays a role in alcohol and tobacco use is useful for gaining an insight into the profile of OSY at risk for alcohol and tobacco use. Registration of OSY youth and community-based peer led programmes that include influential OSY family and friends could be beneficial.

## Introduction

Tobacco and heavy use of alcohol result in millions of deaths annually, with the majority of tobacco and alcohol related deaths occurring in low- and middle-income countries (LMICs) [1]. Alcohol and tobacco use, like in many other LMIC's is prevalent among adolescents [2, 3]. In South Africa, categorised as an LMIC, national studies found that past month alcohol (54.7%) and tobacco use (50.4%) were significantly higher among those who dropped out of school [4, 5] compared to school-going learners who reported 35% and 21% alcohol and tobacco use, respectively [3]. These risk behaviours are associated with poor educational outcomes, [6] diseases, morbidity, and mortality [7] which is a major public health concern.

**Funding:** Rachana Desai receives salary support from a University of Witwatersrand DST-NRF Centre of Excellence in Human Development Postdoctoral fellowship (F14/25 (Desai)) and from the BEACON Cohort, a Welcome Trust Intermediate Fellowship project (211374/Z/18/Z).

**Competing interests:** The authors have declared that no competing interests exist. The authors alone are responsible for the content and writing of this paper. This does not alter our adherence to PLOS ONE policies on sharing data and materials.

Previous South African studies have mainly focused on school-going learners and their alcohol and tobacco use [8, 9]; however, those who drop out of school in South Africa have received less attention. According to the latest UNESCO report globally, 258 million children and adolescents (ages 7–19 years) had either never started or dropped out of school, and more than half (58%) of these youth were living in sub-Saharan Africa [10]. In South Africa, only 52% of the age-appropriate population remained in school until the last grade of high school in 2016 [11, 12]. South African studies show that reasons for leaving school include poverty [13–15], high use of substances [5, 13, 14, 16, 17], bullying [18], boredom [17, 19], family needs (helping support the families, being pregnant, traditional family role expectations) [13, 14, 20], illness [14, 21], disability [14], community violence [14], and school related factors (academic performance, disliking school, not getting along with teachers, being too old for school and disciplinary consequences) [13, 14, 21, 22]. Negative social, health and economic consequences such as unemployment, substance use, delinquency, and poor mental and physical health are usually associated with early school leaving [5, 13, 16]. Given the high prevalence of alcohol and tobacco use among OSY in South Africa, it would be useful to understand the determinants of alcohol and tobacco use in this population.

To our knowledge, no previous studies have examined social network determinants among OSY regarding tobacco and alcohol use. One South African study found that leisure motivation and leisure boredom may be associated with substance use among school dropouts [17], which may extend to the characteristics of the social network of OSY. Previous studies including school-going adolescents showed that drinking and smoking behaviour tend to be modelled after friends' drinking [23–26] and smoking behaviour [27–31]. Partners [32–35] and immediate family (siblings and parents) [36–40] were also significantly associated with adolescent tobacco and alcohol use. However, these studies only considered adolescents attending school who spent a substantial amount of time with school-going peers and teachers. OSY do not have the protective factor of schools, such as the supervision and positive mentoring of teachers and peers, and are more vulnerable to the experimentation and uptake of alcohol and tobacco use [16, 41–44]. Our study will be the first to explore the composition of OSY's social network, and how these social networks support or constrain OSY's alcohol and tobacco use.

Numerous theoretical frameworks have been used to explain the processes by which interpersonal relationships may influence an individual's health-risk behaviour. Adolescence is a transitional period during which peers gain more importance as a means of developing a sense of belonging, self-concept, and support [45]. In the context of alcohol and tobacco use among OSY, the social learning theory [46], the social identity theory [47] and the social network theory [48] provide frameworks for understanding the individual's interpersonal relationships and their cognitions concerning the larger social system [49]. Social networks refer to the connections and nature of interactions between individuals in a social system, which may facilitate the uptake and spread of resources and behaviours [26, 40, 50]. The most distinguishing feature of the social network theory is its two-fold focus on both the individual actors and the social relationships connecting them [50], which can facilitate or inhibit behaviour. The social learning theory considers the acquisition and continuation of behaviour [51]. Adolescents are likely to imitate those with whom they have the greatest amount of contact and continue behaviour based on the rewards and punishments [49]. The social identity theory proposes that a portion of the self-concept of individuals is dependent on the normative values and behaviours of groups that they belong to [47]. Similarity, smoking and drinking behaviour may be explained by the social identity theory whereby individuals tend to act following group norms, adopting them as their own [49]. These theories will be considered in this study to understand the role that interpersonal relationships play in alcohol and tobacco use among OSY.

Understanding the adolescent's social network may be useful to identify the profile of OSY that are most vulnerable to alcohol and tobacco use. Although the WHO defines adolescents as individuals between ages 10–19 years [52], this study focuses on adolescents between 13–20 years as grade repetition in high school is high [53]. Also, a slightly older sample of adolescents was targeted because dropout tends to increase from the age of 15 (grade 9) [5]. National studies show that alcohol and tobacco are the most prevalent among adolescents compared to other addictive behaviours such as illegal and other drug use [2, 3]. Moreover, alcohol and tobacco use is usually initiated between 12–14 years and is highly prevalent in the slightly older age group [3]. To our knowledge, no qualitative study has been conducted, specifically focusing on 13-20-year-old dropouts in South Africa. Qualitative studies may assist in acquiring a deeper understanding of the composition of OSY social networks and interactions as determinants of alcohol and tobacco use among OSY. Therefore, this study aims to qualitatively understand the composition of OSY social networks and explore how these social network relationships and interactions facilitate or constrain alcohol and tobacco use.

## Materials and methods

### Study setting

This study was conducted in the Western Cape, the fourth largest province in South Africa within an urban district characterised by high rates of school dropout [54]. Schooling is compulsory for all South African children from the age of 6 years (grade 1) to the age of 15 years (grade 9). Primary education consists of two phases: the foundation phase, and the intermediate phase. All school governing boards of public schools must supplement government funding by charging school fees and doing other reasonable forms of fund-raising. Caregivers who cannot afford to pay school fees may apply to the school governing board for conditional, partial, or full exemption from paying school fees. Eligibility for full and partial school fee exemptions is calculated based on parental income in relation to the fees [55]. The right not to charge school fees is limited to the schools that have been declared 'no fee schools,' which is based on the economic level of the community around the school. From grade 9, however, children do not benefit from the no-fee policy. Although we did not ask respondents the school they previously attended, they were recruited from urban areas that comprised of both no paying and fee-paying schools.

### Respondents and sampling

Adolescents between the ages of 13–20 years who were not currently enrolled in secondary or high school for that academic year and have not completed their secondary or high schooling were eligible to participate in the study. Those who were enrolled in college or vocational training were excluded. Eligibility was further confirmed through contacts that the potential participant provided. Because there was no available register of OSY, respondents were recruited using respondent-driven sampling (RDS) [56]. Using RDS, the initial sample or "seeds" of OSY were purposefully obtained by data collectors through a community youth group and approaching young people who appeared to meet the pre-determined criteria in selected communities. Initially, eight seeds (4 smokers and 4 non-smokers) were obtained, and they were required to identify up to two other OSY. These respondents recruited by the seeds formed the "first wave" of sampling and were themselves asked to identify and refer a further two more school dropouts (Fig 1). Up to two waves of recruitment were conducted. The seeds consisted of smokers and non-smokers because the initial focus of the paper was on tobacco use among OSY. However, the paper evolved to focus on both alcohol and tobacco use. Despite targeting smokers and non-smokers as the initial seeds, the researchers still acquired a sample that

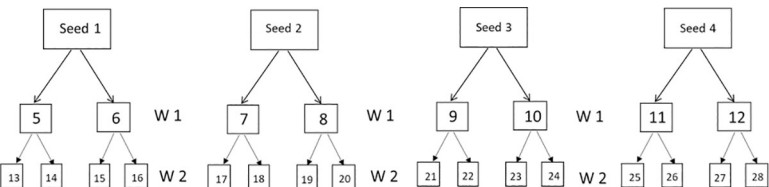

**Fig 1. Respondent-driven sampling for out of school youth–a schematic representation of four seeds.** W 1: wave 1, W 2: wave 2.

contains almost an even number of alcohol and tobacco users while minimising potential bias being introduced, due to the nature of the respondent driven sampling strategy.

## Data collection

The interviews were conducted in-person as well as using text messaging on the cellular phone application WhatsApp. Studies have shown that adolescents prefer sharing information and communicating via instant messaging (IM) [57, 58]. Online interviewing allows access to the voices and experiences of "hidden" populations [59–62], that are difficult for researchers to access due to the lack of a sampling frame and geographical location [63]. Compared to the traditional face-to-face interviewing method, qualitative online interviewing saves cost and time, and is convenient for both the researcher and the participant [57, 64]. In the case of text based online interviewing, the use of emoticons and internet slang abbreviations such as "LOL" are commonly used to convey emotion, and has been used previously by interviewers to build rapport [64, 65]. One study systematically compared face-to-face interviewing techniques to instant messaging interviewing, mediated by a computer [57]. Despite taking longer and producing fewer words in the online condition, data quality was unaffected by the mode of data collection (online versus face-to-face) with no differences in the number, depth and type of themes discussed [57]. This suggests that online data collection using cell phone mediated IM, namely WhatsApp, may be a novel, appropriate and feasible method for obtaining sensitive information from school dropouts. Testing of this latter assumption was not part of the present study.

The data collectors recruited and conducted the interviews in the respondents' language of preference (English, Afrikaans or IsiXhosa). In-person interviews were conducted at private locations agreed to by the respondent and interviewer, lasting approximately 30 minutes. Interviews conducted using WhatsApp took approximately 3–5 days on average with one hour of chat time a day. Respondents who completed the interview received monetary compensation (50 ZAR) for their time. Also, monetary incentives (20 ZAR) were provided for every respondent who successfully recruited more respondents who met the eligibility criteria and participated in the study. Of the respondents that were approached, one respondent was unavailable to do the interview and another did not meet the criteria of being between the ages of 13–20 years. Although the target was 56 respondents, data saturation was achieved after 41 respondents, as by that number no new views were emanating from the interviews since the last three respondents. Of those, 12 respondents conducted the interview over a cellular phone application, WhatsApp, and 29 respondents conducted the interview face-to-face. The transcripts were translated by the data collectors to English from Afrikaans or isiXhosa and then back translated to check for consistency and correct translation.

## Data collection tools

Open-ended questions with probes were used to guide the individual in-depth interviews. All authors of the study designed the questions through a reflective, iterative, and dialogic process.

Respondents were asked to describe their relationships with their family, friends, and other important people in their lives to gain a sense of their social network. They were further asked to discuss their own alcohol and tobacco use, and alcohol and tobacco use of their social network, as well as to elaborate on how important social network members played a role in their alcohol and tobacco use. The semi-structured discussion guide was designed in English and translated into isiXhosa and Afrikaans. The English version is available as a S1 File.

## Analysis

Atlas.ti version 8 was used to code and analyse the data using a content analysis approach [66]. The six-phase approach by Braun and Clarke (2018) was followed to code and analyse the interview data. First, the first author read the transcripts several times and noted initial ideas. Secondly, two independent researchers (RD and AM) blindly coded four transcripts, and preliminary codes were developed and defined based on the objectives of the study. The two researchers compared and discussed the coding until consensus was reached. A final codebook was developed, and the remaining transcripts were coded. The final codebook was reviewed by all authors. Thirdly, the coded transcripts were analysed by running query reports and primary document tables of emerging themes. The fourth phase included review and refining of the themes and sub-themes by all authors. In phase five, the themes and sub-themes were named, and clear working definitions were provided for each theme. Lastly, extracts were selected, and the results of the analyses were organised and presented in this manuscript.

## Ethics and consent

Ethics approval was obtained from the Human Sciences Research Council (Protocol number: REC 2/23/08/17). Youth who are out of school, maybe out of school without their caregiver's knowledge or have unstable family support structures. The nature of the research included minors who might have been unwilling to participate if they had to divulge the nature of the research to their parents or caregivers to obtain parental permission. Given the low risk of the study on respondents, obtaining independent consent from the youth respondents themselves was feasible and increased participation. Therefore, in line with the South African National Department of Health Ethics guidelines (2015) section 3.2.2.4., permission for independent consent for minors was obtained [67]. Once eligibility for participation was established, full written informed consent was obtained from each respondent. All names in the transcripts and extracts presented below were removed or replaced with pseudonyms.

## Results

### Characteristics of the sample

A total of 41 OSY participated in the study. The study had slightly more females (n = 22) than males (n = 19). The average age of respondents was 18 years. At the time of the interview, 31 respondents reported using tobacco, 23 respondents reported drinking alcohol, 10 did not use tobacco, and 18 reported not drinking (Table 1). Using either tobacco or alcohol was similar across gender. However, using both alcohol and tobacco was more common among males than females. Those who drank reported drinking during special occasions, the weekend and payday. Those who smoked reported smoking daily, and the frequency of smoking increased on weekends.

### The social network of OSY

Respondents reported that their social network usually consisted of their friends (in-school and out of school friends and romantic partners) and family (parents, siblings, and extended

**Table 1. Behavioural characteristics of the OSY sample.**

|  |  |  | Male |  | Female |  |
|---|---|---|---|---|---|---|
| Characteristics | % | n | % | n | % | n |
| Total | 100 | 41 | 46 | 19 | 54 | 22 |
| Tobacco users | 65 | 31 | 55 | 17 | 45 | 14 |
| Non-tobacco users | 34 | 10 | 20 | 2 | 80 | 8 |
| Alcohol users | 56 | 23 | 48 | 11 | 52 | 12 |
| Non-alcohol users | 43 | 18 | 44 | 8 | 56 | 10 |
| Alcohol and tobacco users | 39 | 16 | 63 | 10 | 38 | 6 |
| Non- alcohol and non-tobacco users | 12 | 5 | 20 | 1 | 80 | 4 |

family). As seen in the extracts of conversations between the interviewer (I) and respondent (R), each of these network members seemed to play a role in a respondent's inclination and frequency to smoke and/or drink alcohol among OSY users and non-users of alcohol and tobacco. The following paragraphs demonstrate the role that these different individuals play on OSY alcohol and tobacco use.

## Friends

OSY friends played a major role in their alcohol and tobacco use. OSY claimed to have met their friends from when they were attending school or from residing in the same neighbourhood. OSY claimed to have friends who were both in and out of school.

**Initiation of alcohol and tobacco.** Most respondents who were smoking and drinking at the time of the interview initiated and continued to use alcohol and tobacco in the company of friends who smoked and drank. To strengthen bonds between friends, respondents would increase their alcohol and tobacco frequency by adjusting their behaviour to their friends. Moreover, friends would place direct pressure on respondents to smoke and drink by offering alcohol or cigarettes, show respondents how to smoke, and coerce respondents to smoke or drink alcohol.

"I: What made you start drinking and smoking tobacco?

R: I wanted to know the feeling they are getting when they are drunk and smoking tobacco. Sometimes you want to please your friends and be in the same vibe. When your friends are smoking and drinking then you cannot say no, that's what happened to me." (Female respondent ID 31, 20 years old)

Often respondent's initiation with alcohol and tobacco was driven by curiosity and experimentation. Respondents began experimenting with alcohol and tobacco while they were in school with their school friends and continued to drink and/or use tobacco once they left school. Once leaving school, OSY claimed to drink and/or smoke more frequently compared to when they were in school due to having more free time, being unsupervised and being addicted. Despite national policy prohibiting the sale of cigarettes to minors under 18 years, OSY were able to purchase cigarettes from the shop on their transport routes.

"The first time that I smoked was in grade 10 in Johannesburg, it was me and my friend. I had to take her to the taxi rank every afternoon. We just tried to feel what it felt like and every afternoon we bought a cigarette and just breathed out the smoke. We did not inhale at that time and then I came back to Cape Town and became use to smoking and started smoking that I am still smoking today." (Female respondent ID 13, 19 years old)

"I: How old were you when you first started to use tobacco and drink alcohol?

R: When I was 13 years and I was doing Grade 8

I: With whom?

R: With my primary school friends, we were together at high school even." (Male respondent ID 39, 20 years)

OSY who were non-users claimed to have experimented with alcohol or tobacco in the presence of their friends. The experience of the unpleasant physiological and emotional feelings associated with using tobacco and alcohol played a role in their decision not to continue. Respondents would associate alcohol and tobacco use to other negative risky behaviours such as violence, sex, and bad company. Although OSY would initially adjust their behaviour to that of their friends, eventually OSY would develop strong opinions on the negative consequences of using tobacco and alcohol from their own experiences.

R: Yes, I tried smoking, consuming alcohol, smoking marijuana, and hookah pipe.

I: Okay, so you have tried everything? Can you tell me about the first time that you smoked and consumed alcohol or tried to use it?

R: I wasn't the same person that I am now. That was when I tried to do everything friends do. Sitting at a shebeen, walking around in the night and became friends with older men.

I: And you feel that it was not for you?

R: No, it was definitely not for me." (Female respondent ID12, 19 years old)

R: ". . .I don't consume alcohol. So, I don't know what is the purpose of consuming alcohol because it makes people violent and most people don't know how to conduct themselves when they have consumed alcohol because they abuse it. So, in my opinion alcohol has a bad influence on people and that is why I don't consume alcohol." (Female respondent ID 13, 19 years old)

**Financial resources.** Friendship groups provided a support structure and social opportunities which would facilitate alcohol and/or tobacco use among OSY. Cited commonly among the males of this sample, the amount of alcohol and tobacco consumed was dependent on the number of friends and the amount of finances available. Each member of the friend group would contribute financially towards purchasing cigarettes and alcohol, and this would be shared amongst them. Purchasing alcohol and cigarettes in bulk made it more affordable. Thus, those who did not have the financial resources had their alcohol or cigarettes purchases partially or fully subsidised by their friends. Given national regulations prohibiting the sale of alcohol and tobacco to minors, older OSY may have also purchased the alcohol and tobacco on behalf of the group.

I: "How many drinks do you have in one week?

R: It depends what we have on the table and how many people are drinking at that moment because we are four in my group. We usually contribute with a R150 each person.

I: In that amount of money what do you buy?

R: It's a bottle of brandy 750ml, wine and few of 6 dozen of cans of beer/ciders." (Male respondent ID 37, 19 years old).

R: We are coming from different backgrounds and our homes are not the same. For example, we plan to contribute with R150, and it happens that you do not have the whole amount, maybe you have R90 instead of R150. It doesn't mean I cannot go with them. That's the amount you have, at least you came with something" (Male respondent ID 39, 20 years old)

**Leisure boredom.** Given that respondents were not attending school and mostly stayed at home, male participants (n = 6) reported that they were bored in their free time. The participants spent much of their free time during the week at their friends' homes, who were also possibly OSY. Boredom with friends may have inevitably led to tobacco and/or alcohol use as well as contributed to their addiction. The quotations below suggest that OSY youth and their OSY friends would smoke and/drink due to boredom, they had more unstructured time and unsupervised opportunities to use tobacco and or alcohol.

"I: And what do you do every day to keep yourself busy?

R: Every day I am with my friends.

I: And there, what do you do?

R: We smoke the hookah pipe, maybe walk with the dogs but we are just at home." (Male respondent ID 26, 19 years old)

". . .all of us smoke together. We walk together every day then we smoke marijuana, cigarettes and all of that." (Male respondent ID 8, age unknown)

**Reluctance to alcohol and tobacco.** In three instances, there was a reduction in smoking and drinking among female respondents using alcohol and/or tobacco. This occurred when friends would collectively decide to reduce their alcohol and tobacco use, or when respondents had non-drinking or non-smoking friends, who were usually in school.

"R. . .I think that's also one of the reasons I like to be friends with my old school friends because they are not drinking. I buy [a soft drink] and that's it. When I go with other friends, I buy a carry pack [6 bottles of alcohol], but when I go with other ones I don't drink but I feel happy. . ." (Female respondent 13, 17 years old)

Some of the respondents (n = 8) who reported not using alcohol and/or tobacco still had smoking and/or drinking friends. In the company of their friends who would smoke and/or drink, they would not feel coerced by their friends to change their behaviour, their friends were more accepting of their non-using behaviour and claimed that their friends would not drink or smoke excessively in front of them. Participants would engage in other leisure activities with friends such as going to the mall, attending church, playing sports, or making music. These respondents tend to not change their behaviour to that of their smoking or drinking friends.

"I: maybe smoke more or less or consumes more alcohol or less?

R: Neither one of them.

I: Is it? After you left school?

R: After I left school, neither one of the two. I only chilled with my friends. That is all and even if they smoked, I would not smoke. I will only sit by them and the music is playing that is all." (Male respondent ID 38, 17 years old)

Three female respondents reported that their romantic partners contributed towards their cigarette and alcohol purchases. Of those responses who reported using alcohol and/or tobacco, most of their partners discouraged this behaviour by associating smoking and drinking with being unsuitable and unattractive. Some respondents tended to smoke and drink less in the presence of their partners compared to when smoking and drinking with friends.

"I: Where do you get money for drinking alcohol?

R: I get pocket money from my boyfriend and my father. I don't stay with my boyfriend, but he stays just around the corner. He usually spends his weekends with his friend, and I enjoy myself with my friends too.

I: He doesn't say anything when you are drunk?

R: I don't stay in the same place as him when I am going to drink with my friends because he doesn't want me to drink. He told me when I am drunk, I don't want to listen, I am too noisy and using vulgar language." (Female respondent ID 31, 20 years old)

I: "Ok. What information have you been told about drinking by him?

R: I will end up being something else and being ugly" (Female respondent ID 30, 20 years old)

## Family

The majority of the respondents resided in a household consisting of biological parents, siblings and at least one non-parental member such as an aunt, uncle or grandparents. Family tended to play a facilitating and inhibiting role in OSY alcohol and tobacco use.

Family members of similar ageIn some cases, siblings played a role in the respondents' alcohol and tobacco use. Siblings of a similar age to the respondent were more likely to drink together (n = 6). Some respondents would also consume alcohol with their siblings' friends. Siblings who drank and smoked also contributed financially towards the respondents' alcohol and tobacco use.

"I: Are there are other family members who encourage you to stop drinking?

R: No, but sometimes I do drink with my older brother when he drinks with his friends

I: Can you tell me why you think your brother doesn't mind when you drink with him?

R: There is not much difference in age between us." (Male respondent ID 39, 20 years old)

Extended family also seemed to play a role in facilitating respondents' alcohol and tobacco use. Extended family members of a similar age to the respondents who smoked or drank alcohol, were less likely to discourage cigarette and alcohol use. Some respondents reported that they initiated and used cigarettes with their cousins.

"I: Okay. What information were you given by friends and family regarding smoking and the consumption of alcohol? Is there anyone that gave you information?

R: Yes, my cousin. It is almost like he taught me to smoke. He told me I must just do it once and I coughed. So I told him that I want to do it continuously because he also smokes and that was how he taught me to smoke.

I: And he didn't tell you not to smoke because it is bad for you or nothing like that?

R: No." (Female respondent ID 24, 17 years old)

**Elder family members.**   The majority of older family members discouraged alcohol and tobacco use by highlighting its adverse health effects and social consequences, even though they smoked or consumed alcohol. Most respondents continued to use tobacco or alcohol, even though their family did not support it. Often respondents hid their alcohol and tobacco use from their family or use little to no tobacco or alcohol in the presence of their older family members out of respect or fear.

". . .My grandmother always told me that my grandfather use to smoke and it was not easy for him to stop. The day when he stopped smoking was the day he died. So, you better stop now before it is too late. My grandmother also said that I think that it is easy to stop but it is not. She also told me about the dangers of smoking that my lungs will collapse. But then I just tell myself, arg to hell old women. I am at that stage that when I want to smoke, I want to smoke." (Female respondent ID 13, 19 years old)

In one case, a non-user of alcohol and tobacco reported residing with a sibling who was older and took on the role of the caregiver. In this case, the sibling played a role in the respondent's decision not to consume alcohol after being reprimanded for initiating alcohol use.

R: "My sister is happy that I am no longer drinking. I remember on the day after my birthday she was mad with me. She didn't hit me because she respects my special day, but she warned me. All I can say she was not happy with my decision" (Female respondent ID 28, 17 years old).

**Lack of parental support.**   Respondents highlighted that their relationship with parents affected their alcohol and tobacco use. Some respondents attributed their alcohol and tobacco use to not getting along with parents, feelings of neglect, grief and coping with household issues. Those neglected by their parents attributed their alcohol and tobacco use to the stress of contributing towards the household income.

"R. . .Cause the stress levels went higher from like my house problems and my mother is also sick, she like has a heart problem, she has a hole in the heart. And another thing, uhm, we like, my whole family, my mother and my sisters, my two sisters and my dad, we're not into talking a lot. We like don't talk a lot. We talk, but we like, how can I say that, family time is not there, how can I say that, you see family time is not there, to put it in that way. Sometimes, with me it's like, my family is not really away from me but sometimes I feel so, then I go smoke. . ." (Male respondent ID 25, 18 years old)

"I: What made you start drinking?

"R: It's a frustration of losing parents. My mother passed away and I don't know my father" (Female respondent ID 27, 17 years old)

"R: I find it difficult to stop [drinking] because I'm staying alone and I am always thinking about food and clothes to wear, nothing else." (Female respondent ID 34, age unknown)

**Household norms.**   Respondents who are non-users of tobacco and alcohol use reported lower instances of parental smoking and drinking. Although these OSY had some family members who smoke and or drink, these behaviours were seen to be controlled, away from the home, out of sight or only done on special occasions.

"R: Only my father smokes but he never smokes in the house. He always smokes outside. And they don't drink. It will only happen occasionally maybe just wine or so, but they won't drink in front of us." (Male respondent ID 38, 17 years old).

Moreover, it was found that among non-users of alcohol and tobacco (n = 3), families with religious and traditional affiliations were found not to engage in smoking or drinking and tended to condone these behaviours, which also extended to elders in the extended family

I: "What information have you been told about smoking and drinking by your family or friends?

R: I must not drink again because we are not the family who is drinking. We are the Christians and I was not raised to be the drinker. Even my uncle is not a person who's drinking but only on special occasions, like on ceremonies. He is not supporting negatives things and he is very traditional" (Female respondent ID 28, 17 years old)

## Discussion

This study aimed to understand how social network determinants play a role in alcohol and tobacco use behaviours in a group of OSY in South Africa. Overall, the social environment, which included friends and family, played a facilitating or inhibiting role in OSY's alcohol and tobacco use. The findings of this study have provided valuable insight into the profile of OSY at risk for alcohol and tobacco use.

The findings from this study suggest the value of examining the friendship group of OSY as a determinant of OSY alcohol and tobacco use. Similar to studies conducted among in-school learners, OSY would imitate their friend's behaviour concerning alcohol and tobacco use, and continue to use it with their friends [31, 49, 50, 68]. In addition, OSY who were current tobacco and alcohol users tended to spend more time with other OSY friends due to boredom, offering more opportunities and unsupervised time to engage in risk behaviours, rather than spending that time in school. A previous study also found that OSY tend to have more OSY friends [69] and that the smoking behaviour of OSY is associated with that of their OSY friends [70]. In a few cases, being in the company of non-smoking and non-drinking friends or romantic partners would reduce OSY's alcohol and tobacco use. These findings suggest that OSY tobacco and alcohol use is a learned behaviour with reinforcement in the peer context [71].

In this study, OSY and their friends often drank and smoked in groups, conveying a sense of collective identity and group membership [47]. Furthermore, each group member contributed to the purchase of cigarettes and alcohol. Those who could not contribute financially were assisted by their friends or partners and were thus able to continue using cigarettes and alcohol. The national policy states that tobacco and alcohol may not be purchased by minors under 18 years [72, 73]. Older OSY may have also purchased the alcohol and tobacco for underage OSY. The availability of these resources may explain how female dropouts experiencing financial difficulties had access to cigarettes, as seen in a previous study by Desai and colleagues [5]. Enforcement of national policy prohibiting the sale of alcohol and tobacco to minors or raising the minimum age of tobacco and alcohol purchases may be beneficial. Those OSY who were not users of alcohol or tobacco reported having friends who engaged in these risk behaviours but would not play a role in their decision not to smoke and/or drink, possibly due to individual characteristics or the characteristics of their friends.

The family social network comprising of parents, siblings, and extended family, was also a determinant of alcohol and tobacco use among OSY in this study. OSY mostly resided with parents and at least one non-parental adult, which included grandparents, aunts, uncles,

cousins, and other relatives. Interestingly, respondents described their home as consisting of backyard dwellings where the extended family or the children resided. Backyard dwellings are informal shacks, typically erected in the yards of other properties, and is unique to South Africa [74]. The extended family plays a crucial social safety net in sub-Saharan Africa [75]. In cases where a child loses their parents or when parents do not have the resources to support the child, it is common for the child to reside with their relatives [75, 76]. Our study also found that older family members such as parents, aunts, uncles and grandparents and siblings would discourage alcohol and tobacco use, even if they used alcohol or tobacco. However, those OSY who were smoking, or drinking were not affected by these discouragements. This finding may be attributed to members in the household smoking and drinking, the lack of in-home smoking rules, lack of religious norms, poor quality and frequency of caregiver-child communication, as well as the lack of value OSY place on their caregivers' opinions about smoking and drinking [36, 77, 78]. This calls for further exploration on how to incorporate broader caregiving strategies into alcohol and tobacco prevention interventions among OSY.

Some OSY in this study attributed their alcohol and tobacco use to coping with household issues such as not being able to get along with parents, the stress of contributing towards the household income, feelings of neglect by parents or the loss of a parent. Moreover, previous studies have found that living in single-parent families, step-parent families or no-parent families are all associated with higher odds of ever/lifetime smoking and drinking [79–82]. Many children in South Africa grow up in fractured families usually without one or both biological parents. These families are also faced with poverty and unemployment [15, 76]. Studies have shown that socioeconomic disadvantaged adolescents are more likely to take up tobacco and alcohol use [83, 84]. Similarly, leaving school due to financial difficulties and lack of family support was associated with alcohol and tobacco use among OSY in South Africa, but the strength and direction of these associations were dependent on gender and geographical area [4, 5]. Currently, South African policies exempting students who do not have the affordability from paying tuition fees should be enacted [85]. Coping mechanisms and strategies can also be incorporated into alcohol and tobacco prevention interventions among OSY.

To our knowledge, this was the first study that explored the social network determinants of tobacco and alcohol use among OSY in South Africa. A limitation of the study was that the views expressed in the interviews are more representative of an older sample of OSY (average age 18 years) who are legally permitted to purchase tobacco and alcohol in South Africa. More research is needed among younger OSY between ages 13–15. The different interviewer methods may have influenced participant responses. However similar a previous study [57], the findings of this study suggest that data quality is unaffected by the mode of data collection. The findings of this study may not be unique to OSY and therefore the study may need to be replicated among in-school learners to draw further comparisons and conclusions with older and younger OSY. We acknowledge that sections of data may be included in multiple themes with some overlap between themes. However, the researchers considered how each theme fit into the overall story about the entire data set. This study does however provide some insight into the unique social network determinants of alcohol and tobacco use among OSY, which are confirmed by aspects of previous studies among OSY and in-school learners. Furthermore, due to OSY being a hard-to-reach population, transcripts were not returned to respondents for comment and/or correction due to difficulty in contacting them again.

## Conclusions and implications

While similarities in social network determinants exist with in-school learners, this study additionally found that facilitators of alcohol and tobacco use include having more access to

financial resources from peers and OSY having more OSY friends who may use alcohol and tobacco. OSY's experience of the lack of parent support, coping with the loss of a parent and the larger role that extended family may play were also facilitators of alcohol and tobacco among OSY.

The findings of this study have important implications for alcohol and tobacco prevention and cessation programs among OSY. Attempts should be made to register all those who drop out of school to allow for tracking of school dropouts for intervention. Given that OSY do not have the protective factor of school-based interventions, interventions appropriate and accessible through the community is warranted. Furthermore, peer led programmes which have been shown to be successful among in-school learners [86, 87] could include a peer led OSY component in that encourage the diffusion of non-smoking and non-drinking norms in the community.

## Supporting information

**S1 File.**
(DOCX)

**S1 Matrix code.**
(XLSX)

## Author Contributions

**Conceptualization:** Rachana Desai, Robert A. C. Ruiter, Priscilla S. Reddy, Liesbeth A. G. Mercken.

**Data curation:** Rachana Desai.

**Formal analysis:** Rachana Desai, Ansuyah Magan, Liesbeth A. G. Mercken.

**Funding acquisition:** Rachana Desai, Robert A. C. Ruiter.

**Investigation:** Rachana Desai, Liesbeth A. G. Mercken.

**Methodology:** Rachana Desai, Robert A. C. Ruiter, Liesbeth A. G. Mercken.

**Project administration:** Rachana Desai, Robert A. C. Ruiter.

**Resources:** Rachana Desai, Priscilla S. Reddy.

**Software:** Rachana Desai.

**Supervision:** Robert A. C. Ruiter, Priscilla S. Reddy, Liesbeth A. G. Mercken.

**Validation:** Rachana Desai, Robert A. C. Ruiter, Ansuyah Magan, Liesbeth A. G. Mercken.

**Visualization:** Rachana Desai.

**Writing – original draft:** Rachana Desai, Robert A. C. Ruiter.

**Writing – review & editing:** Rachana Desai, Robert A. C. Ruiter, Ansuyah Magan, Priscilla S. Reddy, Liesbeth A. G. Mercken.

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
