## [Decision Letter · Decision Letter 0]

2 Jul 2020

PONE-D-20-17565

Social network determinants of alcohol and tobacco use: a qualitative study among out of school youth in South Africa

PLOS ONE

Dear Dr. Desai,

Thank you for submitting your manuscript to PLOS ONE. After careful consideration, we feel that it has merit but does not fully meet PLOS ONE’s publication criteria as it currently stands. Therefore, we invite you to submit a revised version of the manuscript that addresses the points raised during the review process.

In particular, please pay attention to 1) the rationale (why the focus on tobacco and alcohol?), 2) presentational issues (further details are required in your methods section), 3) depth of findings (in your results section, more quotes or longer quotes are needed to support your conclusions) and 4) interpretation (additional limitations need to be considered; what does this study add?).

We look forward to receiving your revised manuscript.

Kind regards,

Lion Shahab, MA MSc MSc PhD CPsychol

Academic Editor

PLOS ONE

Journal Requirements:

2. Please include additional information regarding the interview guide or script used in the study and ensure that you have provided sufficient details that others could replicate the analyses.

For instance, if you developed a guide as part of this study and it is not under a copyright more restrictive than CC-BY, please include a copy, in both the original language and English, as Supporting Information.

'We would like to acknowledge the scholarship of the Foundation Study Fund for South African students in the Netherlands.'

'The authors received no specific funding for this work.'

Reviewers' comments:

Reviewer's Responses to Questions

**Comments to the Author**

1. Is the manuscript technically sound, and do the data support the conclusions?

Reviewer #1: Partly

Reviewer #2: Partly

2. Has the statistical analysis been performed appropriately and rigorously? 

Reviewer #1: N/A

Reviewer #2: N/A

3. Have the authors made all data underlying the findings in their manuscript fully available?

Reviewer #1: No

Reviewer #2: Yes

4. Is the manuscript presented in an intelligible fashion and written in standard English?

Reviewer #1: Yes

Reviewer #2: Yes

5. Review Comments to the Author

Reviewer #1: Abstract

1.Lines 28-30: "Interventions preventing the use of alcohol and tobacco may benefit from considering the family and friend social networks of OSY.” There is little mention of the development of interventions based on the findings of this study elsewhere in the manuscript. The authors may consider expanding upon this if it is to be included in the abstract. What kind of interventions? How would considering these factors help prevent alcohol and tobacco use? Why is it important to target OSY?

Introduction

1.Please provide some more detailed background about the schooling system in south Africa (public schools are fee paying/subsidised, free for low-income etc), and the system that most/all participants were recruited from. This will help frame the sample population for the reader.

2.Lines 37-39: Are there any estimates of prevalence other than past-month alcohol and tobacco use? For instance, weekly or daily use? The past month indicator will capture experimentation as well as more frequent usage, so it would be helpful to have another indicator of more frequent usage (if available!).

3.Lines 42-53: What is the socio-economic patterning of OSY? Presume it is largely driven by those who suffer greater disadvantage?

5.Lines 49-51: Might delinquency, poor mental health, and poor physical health also influence the likelihood of dropping out of school (i.e. some reverse causality?)

Materials and methods

1.The inclusion/exclusion criteria for participants in the study could be made clearer.

2.Lines 108-109: Please provide further details about the 4 smoker and 4 non-smoker sampling strategy, what is the rationale for this and how does this apply to alcohol use?

3.Lines 118-120: Why was WhatsApp used as a data collection tool in addition to in person interviews? Has this been done before and is it a valid method to collect this information? Why was telephone interviewing not considered? It may be a more flexible approach but it does it allow for skill of the interviewer to elicit deeper and more revealing responses?

4.Lines 122-124: How verify that individual was not in school? Monetary incentive might encourage recruitment of ineligible participants?

5.Why is qualitative methodology useful here? If the aim was to identify the composition of the social network, and how this influenced tobacco/alcohol use (e.g asking whether friends/family make the individual less likely/more likely to use product) might a simple survey have been more appropriate and less resource intensive?

Data collection

1.How many interviews were in person and how many were conducted via text message?

2.As above, is using text messages a valid methodology? It may be flexible but do the responses warrant inclusion with more in-depth interviews? Might people using WhatsApp to respond be qualitatively different to those who agreed to have in-person interviews?

Results

1.A summary table including information on the characteristics of the sample is necessary. While this is not a quantitative analysis, it would still be helpful for the reader to get an understanding of the participants. This could include the age range, gender, the number that were smokers/drinkers, how long they've been out of school etc)

2.Please give an indication of how many respondents are being referred to, rather than ‘some’. While quantifying the results may not be the goal of this qualitative research, it is helpful to know how many participants are represented in each result, especially in a large (n=41) qualitative sample.

3.Was any attempt made to explore the different ways in which tobacco or alcohol is used by OSY? Is there more frequent usage, or more experimentation?

4.Were there any differences between male and female OSY? Or between younger vs older participants?

5.The data appear to be lacking in depth. Most of the example quotes appear limited/short, and were not explored further. Might this reflect the WhatsApp interview technique? The data presented are do not currently seem to be sufficient to represent/support the findings.

6.Lines 149-152 in the analysis section mention the generation of themes and sub-themes. It is not clear what these themes are as they are not specifically mentioned in the results or discussion. The results are grouped under different members of the social network – are these the themes? Some clearer structure to the results, and corresponding discussion is needed.

7.The quotes could do with an anonymous identifier. Otherwise it is unclear whether the quotes used came from a wide range of participants or from a select few who gave more coherent responses. E.g. participant 1, female, age 19 or participant 2, male, age 20 etc…

A coding matrix in the supplementary materials would also be helpful. This could show what the collection of quotes were that generated the themes in this study. The method clearly outlines the process to reach each theme, but without some more examples of what content actually created them, it is difficult for the reader to get an idea about how valid the themes are. The form submitted with the manuscript mention that data is readily available, but it does not seem to have been submitted with the manuscript for review?

8.As above, it would be helpful to see the interview discussion/topic guide that was used.

9.Is the sample representative of younger OSY? What is the range of ages?

10.Line 184-186: Friends either played a facilitating or inhibiting role in the tobacco and alcohol use of current OSY users, but did not play a role among non-users of alcohol and/or tobacco.” This discrepant finding is interesting but requires more elaboration. What was the explanation as to why smoking/drinking friends did they not play a role initiation or use among these non-users?

11.Lines 213-214: The word ‘affordability’ does not make sense in this context. Affordability refers to the cost of something or its inexpensiveness. Perhaps ‘financial resources’ can be used instead?

Discussion

1.Lines 364-365: What about the participants who drank/smoke despite their partner disapproving? There is no discussion of this apparently discrepant finding.

2.Lines 390-392 This is the first time that religiosity is mentioned in the manuscript. If it is a key theme then it should be referred to/exemplified in the results section, before being discussed later on.

3.Lines: 408-412: Additional limitations need to be considered.

-It is possible that the findings in this study are not unique to OSY, but also reflect youth who attend school.

-Representativeness is mentioned as a limitation due to the geographic region, but does this not also apply to younger OSY? If there were no/few younger members in the sample then these results may reflect an older demographic (average age is 18, which in terms of ability to purchase substances is an important difference.)

-There is no mention of reflexivity of the researchers. How might this have influenced the conduct and analysis of this study? Has the relationship between researcher and participants been adequately considered? It would be helpful to see the researchers critically examine their own role, potential bias and influence during (a) formulation of the research questions (b) data collection, including sample recruitment and choice of location.

-Are there any limitations in using different methods of data collection? Using WhatsApp is convenient but it may limit the depth of responses due to it being a burden to write out a long response.

-Some more time spent outlining the importance of these findings is warranted. What new knowledge has been created? The introduction lines 58-61 outline some findings from research among school attending youth – these appear to be very similar to this study’s current findings/conclusions. What are the differences? This could be clearer in the discussion section.

4. As above - despite there being no comparison with school attendees in this study, how might these results differ compared to previous research?

5.This study focuses on the social networks of OSY, but the wider determinants of smoking/alcohol need to be considered in the discussion. E.g. The availability of cigarettes/alcohol, any wider socio-economically patterns in usage, government policies influencing price/availability.

6.More discussion is warranted on contribution the study makes to existing knowledge or understanding. Use in intervention development is mentioned but what exactly would this look like? What targets have been created for intervention development? Why would an OSY tailored intervention be more beneficial than a more generic intervention targeting all youth (given that the social-network determinants appear to be similar between the two)?

Reviewer #2: In this manuscript, the authors explored through qualitative research social network determinants of alcohol and tobacco use among out of school youth (OSY) in South Africa. They conducted 41 interviews with OSY in a South African urban area and their findings suggest that determinants of alcohol and tobacco use include drinking friends, family members close in age to OSY that drank and smoked and household issues. On the other hand, they found that religiosity, parental control over alcohol and tobacco, romantic partners, and non-smoking or non-drinking friends alleviated alcohol and tobacco use.

The topic is very interesting, but the manuscript needs more attention to the use of English language. Additionally, it could be improved in the following respects:

1)In the abstract the authors mention that ‘an important determinant of alcohol, and tobacco use is the adolescent's social network, which has globally not been explored among out of school youth (OSY). However, the paper only includes participants from South Africa.

2) It is also good practice for qualitative research to state the analytic method in the abstract.

3)The last sentence of the abstract is very general. Authors only look at OSY, so their suggestions for interventions should be relevant to this group of people.

4)Authors should also give a better definition of adolescent and out of school youth regarding the age range. They provide information for different age groups in the Introduction and then they state that their sample included participants age between 13-20 years old. But there is no justification why they chose this age range.

5) Authors present % of tobacco and alcohol use among school going learners and OSY. It would be clearer if they present these figures in the same paragraph.

6) There is no clear explanation why authors are focusing on alcohol and tobacco use and not other addictive behaviours?

7) It is not clear why authors needed 52 participants for their study and if they recruited more than 52. Additionally, why did authors initial recruit 4 smokers and 4 non-smokers and not 2 smokers, 2 non-smokers, 2 drinkers and 2 non-drinkers?

8) How many interviews conducted in person and how many through whatsapp? The authors could perhaps also reflect on whether any differences in quality were observed across those interviewed via the different modalities.

9) Data collection tools would be clearer if they include examples of questions and probs (e.g. including the topic guide as a supplementary file). Did the authors assess participants’ alcohol and tobacco use and how?

10) The authors need to specify who gave them ethical approval.

11) Results section: How many participants use both alcohol and tobacco?

12) It is not clear if the friends of participants are also OSY or not.

13) Results section could benefit with more examples, interview quotations to support the results. Especially results about parents the example provided did not support the findings. The authors also mention that ‘Often respondents hid their alcohol and tobacco use from their family or use little to no tobacco or alcohol in the presence of their older family members out of respect or fear’, but the example provided did not support such findings. It also appears that the analysis is a bit shallow. For example, the theme ‘Friends’ is very broad, and the presented quotation suggests that participants were bored and drinking/smoking because there isn’t much else to do. They also mentioned having resources/money to buy alcohol/tobacco, which is more of an ‘opportunity’ rather than ‘social’ variable that influences use. Again, the theme ‘Parent’ doesn’t quite seem to capture what’s expressed by participants (e.g. drinking due to lack of parental support/grief.

14) Some discussion points are not supported by the findings i.e. ‘Among OSY who were current tobacco and alcohol users, OSY tended to spend more time with other OSY friends’; ‘ These findings may explain why a different sample of females, who dropped out of school due to financial difficulties in South Africa were smoking cigarettes, as seen in a previous study by Desai and colleagues (13)’. In the second example it is not also clear where different refers to?

15) Authors should avoid use language such as substance users when they are only referring to alcohol and tobacco use

16) Discussion about romantic partners is not clear. For example, the point that ‘compared to friends, OSY in this sample would smoke and drink less with their romantic partners, possibly due to spending less time with romantic partners in the context of smoking and drinking’ is not clear how is supported by the results.

17) It would be useful to discuss implications for policy and avenues for future research.

6. PLOS authors have the option to publish the peer review history of their article (what does this mean?). If published, this will include your full peer review and any attached files.

Reviewer #1: No

Reviewer #2: **Yes: **Dimitra Kale

---

## [Author Response · Author response to Decision Letter 0]

16 Aug 2020

We thank the reviewers for these helpful comments, which we have used to strengthen our paper. We respond to each comment below and indicated the changes in track changes in the paper. Reference to lines in the document pertain to the version with no track changes

 We have formatted the paper to meet the journals style requirements

2. Please include additional information regarding the interview guide or script used in the study and ensure that you have provided sufficient details that others could replicate the analyses.

For instance, if you developed a guide as part of this study and it is not under a copyright more restrictive than CC-BY, please include a copy, in both the original language and English, as Supporting Information.

The English version of the interview guide has been provided as Supporting information

The data has been provided as Supporting information titled “coding matrix”. We have requested for the data availability statement to be changed in the revised cover letter.

'We would like to acknowledge the scholarship of the Foundation Study Fund for South African students in the Netherlands.'

'The authors received no specific funding for this work.'

The acknowledgments section has been removed. The authors did not receive specific funding for this work. We have included these changes in the revised cover letter.

Review Comments to the Author

Reviewer #1: Abstract

1.Lines 28-30: "Interventions preventing the use of alcohol and tobacco may benefit from considering the family and friend social networks of OSY.” There is little mention of the development of interventions based on the findings of this study elsewhere in the manuscript. The authors may consider expanding upon this if it is to be included in the abstract. What kind of interventions? How would considering these factors help prevent alcohol and tobacco use? Why is it important to target OSY?

We thank the reviewer for the comment. After revising the manuscript and taking all the comments into consideration, it would appropriate to state the following conclusion in the abstract: “Understanding how the social network of OSY plays a role in alcohol and tobacco use is useful for gaining an insight into profile of OSY at risk for alcohol and tobacco use. Registration of OSY youth and community-based peer led programmes that include influential OSY family and friends could be beneficial. We feel that this conclusion sits closer to the study findings. We feel that this conclusion sits closer to the study findings.

Introduction 

1.Please provide some more detailed background about the schooling system in south Africa (public schools are fee paying/subsidised, free for low-income etc), and the system that most/all participants were recruited from. This will help frame the sample population for the reader.

Thank you for your comment. 

This study was conducted in the Western Cape, the fourth largest province in South Africa within an urban district characterised by high rates of school dropout. Schooling is compulsory for all South African children from the age of 6 (grade 1) to the age of 15 (grade 9). Primary education consists of two phases: the foundation phase; and the intermediate phase. All school governing boards of public schools must supplement government funding, by charging school fees and doing other reasonable forms of fund-raising. Caregivers who cannot afford to pay school fees may apply to the school governing board for conditional, partial, or full exemption from paying school fees. The right not to charge school fees is limited to the schools that have been declared ‘no fee schools,’ which is based on the economic level of the community around the school. From grade 9 however, children do not benefit from the no-fee policy. Eligibility for full and partial school fee exemptions is worked out on the basis of parental income in relation to the fees. 

Although we did not ask respondents the school they previously attended, they were recruited from urban areas that comprised of both no paying and fee-paying schools.

We added a sub-section “study setting” in the methods section to indicate the details of the south African schooling system and the context in which the study took place. We trust that these edits will allow for the paper to read better and provide more context. 

2.Lines 37-39: Are there any estimates of prevalence other than past-month alcohol and tobacco use? For instance, weekly or daily use? The past month indicator will capture experimentation as well as more frequent usage, so it would be helpful to have another indicator of more frequent usage (if available!).

Thank you for the comment. Past month and lifetime alcohol and tobacco use prevalence rates are available for in-school and out of school youth between 13-20 years, which is the age range considered in this study. Although national prevalence of weekly use of alcohol and daily use of tobacco among adolescents aged 15-19 years is available from the South African Demographic Health Survey, it is not clear whether these adolescents are in-school or out of school. We therefore used past month tobacco and alcohol use prevalence rates. Moreover, we will follow the second reviewer’s suggestion of presenting the prevalence of past month tobacco and alcohol use among school going learners and OSY in the same paragraph to allow for clear comparison.

3.Lines 42-53: What is the socio-economic patterning of OSY? Presume it is largely driven by those who suffer greater disadvantage?

We have added more detail on the profile of OSY. The South African literature shows that reasons for leaving school include poverty, high use of substances, bullying, boredom, family needs (helping support the families, being pregnant, traditional family role expectations), illness, disability, community violence, and school related factors (academic performance, disliking school, not getting along with teachers, being too old for school and disciplinary consequences). We trust that his information provides a better understanding of the profile of OSY in South Africa.

5.Lines 49-51: Might delinquency, poor mental health, and poor physical health also influence the likelihood of dropping out of school (i.e. some reverse causality?)

The reverse is certainly true and therefore we changed the phrasing of this sentence to not imply causality. The sentence now reads as “Negative social, health and economic consequences such as unemployment, substance use, delinquency, and poor mental and physical health is associated with early school leaving.”

Materials and methods

1.The inclusion/exclusion criteria for participants in the study could be made clearer.

Adolescents between the ages of 13-20 years who is not currently enrolled in secondary or high school for that academic year and has not completed their secondary or high schooling were eligible. Those who were enrolled in college or vocational training were excluded. We trust that this inclusion and exclusion criteria provides more clarity.

2.Lines 108-109: Please provide further details about the 4 smoker and 4 non-smoker sampling strategy, what is the rationale for this and how does this apply to alcohol use?

The initial focus of the paper was tobacco use among OSY and their social network. As the paper evolved, we focused on both alcohol and tobacco use. Despite targeting smokers and non-smokers as the initial seeds, we still acquired a sample that contains almost an even number of alcohol and tobacco users while minimising potential bias introduced, due to the nature of the respondent driven sampling strategy. 

3.Lines 118-120: Why was WhatsApp used as a data collection tool in addition to in person interviews? Has this been done before and is it a valid method to collect this information? Why was telephone interviewing not considered? It may be a more flexible approach but it does it allow for skill of the interviewer to elicit deeper and more revealing responses?

Given the wide use of cell phones among adolescents, cell phone mediated online communication technologies and instant messaging (IM) services have become popular ways of communication. Studies have shown that compared to other online modalities, adolescents prefer sharing information and communicating via instant messaging (IM) (Lee, 2007; Shapka, Domene, Khan, & Yang, 2016). 

Online interviewing allows access to the voices and experiences of “hidden” populations (Adler & Zarchin, 2002; Ayling & Mewse, 2009; Mathy et al., 2002; Turney & Pocknee, 2005), characterised as socially disadvantaged groups that are difficult for researchers to access due to the lack of a sampling frame and geographical location (Heckathorn, 1997). Compared to the traditional face-to-face interviewing method, qualitative online interviewing saves cost and time, and is convenient for both the researcher and the participant (Jowett et al., 2011; Shapka et al., 2016). 

In the case of text based online interviewing, the use of emoticons and internet slang abbreviations such as “LOL” are commonly used to convey emotion, and has been used previously by interviewers to build rapport (Jowett et al., 2011; Kazmer & Xie, 2008). 

One study systematically compared face-to-face interviewing techniques to instant messaging interviewing, mediated by a computer (Shapka et al., 2016). Shapka et al. (2016) found that despite taking longer and producing fewer words in the online condition, data quality was unaffected by the mode of data collection (online versus face-to-face) with no differences in the number, depth and type of themes discussed. This suggests that online data collection using cell phone mediated IM may be a novel, appropriate and feasible method for obtaining sensitive information from school dropouts. 

Given the novelty of interviewing over WhatsApp, we will be writing a separate paper detailing the equivalence of interview data collected online using text over WhatsApp, with that obtained face-to-face to obtain qualitative data from a hidden population of school dropouts. Given the focus and length of the current paper, the above information was omitted. However, we are open to suggestions on how we should include this information in the manuscript.

4.Lines 122-124: How verify that individual was not in school? Monetary incentive might encourage recruitment of ineligible participants?

Eligibility was further confirmed through contacts that the potential participant provided. This detail has been added to the manuscript. 

5.Why is qualitative methodology useful here? If the aim was to identify the composition of the social network, and how this influenced tobacco/alcohol use (e.g asking whether friends/family make the individual less likely/more likely to use product) might a simple survey have been more appropriate and less resource intensive?

Limited research is known about the composition of OSY social network and the similarities and differences of their social network to in-school learners. Qualitative methods provide a voice to this hidden population and ensures that study findings are grounded in participants' context and experiences. Qualitatively understanding how OSY’s social networks support or constrain alcohol and tobacco use is useful for providing additional validity and contextualization when developing survey instruments.

Data collection

1.How many interviews were in person and how many were conducted via text message?

12 respondents conducted the interview over a cellular phone application WhatsApp, and 29 respondents conducted the interview face-to-face. We added this detail to the manuscript.

2.As above, is using text messages a valid methodology? It may be flexible but do the responses warrant inclusion with more in-depth interviews? Might people using WhatsApp to respond be qualitatively different to those who agreed to have in-person interviews?

Text-based WhatsApp interviews are a valid methodology for the reasons stated above. In the context of the coronavirus, this way of conducting qualitative interviews may offer a good alternative to the traditional face-to-face interviews. 

To minimise bias introduced in the interview conditions, simple randomisation was used to assign the seeds into either the in-person interview condition or WhatsApp interview condition. The seeds further recruited into the interview conditions they were assigned into. If a participant could not conduct the interview in the condition they were assigned into, the alternative interview condition was offered, and they were no longer required to further recruit more participants. Once again, we have omitted this information due to the length and focus of the paper but we are open to suggestions on how to include this information if necessary. 

*Results

1.A summary table including information on the characteristics of the sample is necessary. While this is not a quantitative analysis, it would still be helpful for the reader to get an understanding of the participants. This could include the age range, gender, the number that were smokers/drinkers, how long they've been out of school etc)

We thank the reviewers for the suggestion and included a table of characteristics available to us. Alcohol and tobacco behavioural characteristics were presented across gender. Age was described in the text following the table.

2.Please give an indication of how many respondents are being referred to, rather than ‘some’. While quantifying the results may not be the goal of this qualitative research, it is helpful to know how many participants are represented in each result, especially in a large (n=41) qualitative sample.

We have indicated the number of respondents referred to where possible 

3.Was any attempt made to explore the different ways in which tobacco or alcohol is used by OSY? Is there more frequent usage, or more experimentation?

Respondents were asked to indicate if they used tobacco and or alcohol at the time of the interview, their age of initiation, frequency of smoking and/or drinking per week. Those who did not drink, or smoke were asked about their lifetime drinking or smoking and reasons for quitting. A brief summary description of the frequency of smoking and drinking has been added under the characteristics of the sample. The discussion guide and coding matrix has been included as a supplementary file.

4.Were there any differences between male and female OSY? Or between younger vs older participants?

We unfortunately only had older OSY sample in this study. Using either tobacco or alcohol was similar across gender. However, using both alcohol and tobacco was common among males compared to females. The gender breakdown of alcohol and tobacco use has been added to the sample characteristics table. 

Examining the gender differences was not the main focus of this study however, findings somewhat indicate that being assisted financially by friends and using alcohol and tobacco due to leisure boredom was commonly cited among males. OSY resistance to alcohol and tobacco use and discouragement from romantic partners was mostly cited among the females in this study. We felt that given the small sample size differences found between genders, strong conclusion could not be made between male and female in the study. 

5.The data appear to be lacking in depth. Most of the example quotes appear limited/short, and were not explored further. Might this reflect the WhatsApp interview technique? The data presented are do not currently seem to be sufficient to represent/support the findings.

We have taken the reviewers comments into consideration and added more example quotations to support the findings. We also considered re-ordering the presentation of the quotations, added subthemes where appropriate and explored those quotations in more depth. We trust that the revised presentation of results strengthens the paper.

6.Lines 149-152 in the analysis section mention the generation of themes and sub-themes. It is not clear what these themes are as they are not specifically mentioned in the results or discussion. The results are grouped under different members of the social network – are these the themes? Some clearer structure to the results, and corresponding discussion is needed.

We thank the reviewer for the comment. We have clarified the main themes to be friends and family. The subthemes for friends were initiation of alcohol and tobacco, financial resources, leisure boredom and resistance to tobacco and alcohol. Family subthemes were those of similar age, those elderly, lack of parental support and household norms. The results have been restructured to reflect these themes and sub-themes.

7.The quotes could do with an anonymous identifier. Otherwise it is unclear whether the quotes used came from a wide range of participants or from a select few who gave more coherent responses. E.g. participant 1, female, age 19 or participant 2, male, age 20 etc…

A coding matrix in the supplementary materials would also be helpful. This could show what the collection of quotes were that generated the themes in this study. The method clearly outlines the process to reach each theme, but without some more examples of what content created them, it is difficult for the reader to get an idea about how valid the themes are. The form submitted with the manuscript mention that data is readily available, but it does not seem to have been submitted with the manuscript for review?

We thank the reviewer for the comment. We included an anonymous identifier after each quotation. We have also included a coding matrix with code groups and corresponding quotations, which has been included as a supplementary file. All coded data has been included in this file and will be readily available. We trust that this will reflect the process of how we reached each theme.

8.As above, it would be helpful to see the interview discussion/topic guide that was used.

This has been included as a supplementary file

9.Is the sample representative of younger OSY? What is the range of ages?

Given the average age range of participants was 18 years (SD=1.2), the sample in this study represents older OSY, possibly due to the inclusion of older OSY as initial seeds in the RDS recruitment strategy. The study could have benefited from including seeds from younger OSY groups to get a more representative sample. We have included this in the limitations. 

10.Line 184-186: Friends either played a facilitating or inhibiting role in the tobacco and alcohol use of current OSY users, but did not play a role among non-users of alcohol and/or tobacco.” This discrepant finding is interesting but requires more elaboration. What was the explanation as to why smoking/drinking friends did they not play a role initiation or use among these non-users?

We realise that it would be better to move lines 184-186 to the discussion section as it is more appropriate as a summary of findings about friends. We provide additional quotes and context to smoking/drinking friends not playing a role in the initiation or use among the non-users. We further analysed non-users and it was found that they would engage in other leisure activities such as going to the mall, sports and attending church. They also claim that their friends did not excessively use alcohol and tobacco.

 An in-depth explanation as to why smoking/drinking friends did not play a role initiation or use among these non-users is offered in the discussion section and we trust that this is also sufficient.

11.Lines 213-214: The word ‘affordability’ does not make sense in this context. Affordability refers to the cost of something or its inexpensiveness. Perhaps ‘financial resources’ can be used instead?

We agree with this comment and made the changes in the manuscript

*Discussion

1.Lines 364-365: What about the participants who drank/smoke despite their partner disapproving? There is no discussion of this apparently discrepant finding.

Our results mainly reflected that respondents would drink and smoke less and were discouraged by their romantic partners. We did not find quotations that support OSY who drank and smoked despite their partner disapproving. Given the restructuring of the results section, we now decided to combine the finding pertaining to romantic partners in the previous paragraph with friends. Romantic partners are not the main finding of this study and therefore a full discussion is not warranted. We trust that this presentation of results allows for the paper to read better.

2.Lines 390-392 This is the first time that religiosity is mentioned in the manuscript. If it is a key theme then it should be referred to/exemplified in the results section, before being discussed later on.

We acknowledge that the way religiosity is presented in the discussion section makes this concept seem like a main finding. We have therefore in the results section considered re-ordering the presentation of the quotations, added subthemes where appropriate and explored those quotations in more depth. Religiosity is not necessarily a key theme but it is worth mentioning in the discussion section that lack of religious norms along with lack of in-home smoking rules, poor quality and frequency of caregiver-child communication, as well as how much adolescents value their caregivers’ opinions about smoking and drinking play a role on OSY decision to smoke or drink alcohol (lines 468-471). We trust that this presentation of results and discussion provides more clarity 

3.Lines: 408-412: Additional limitations need to be considered.

-It is possible that the findings in this study are not unique to OSY, but also reflect youth who attend school.

We agree that the findings of this study may not be unique to OSY and therefore the study may need to be replicated among younger and older in-school learners to draw further comparisons and conclusions. This study does however provide some insight into the unique social network determinants of alcohol and tobacco use among OSY, which are confirmed by aspects of previous studies among OSY and in-school learners. We have included this statement in the limitations section.

-Representativeness is mentioned as a limitation due to the geographic region, but does this not also apply to younger OSY? If there were no/few younger members in the sample then these results may reflect an older demographic (average age is 18, which in terms of ability to purchase substances is an important difference.)

We thank the reviewer for this comment and mentioned that the OSY in this sample were representative of an older OSY who are legally permitted to purchase tobacco and alcohol in South Africa.

-There is no mention of reflexivity of the researchers. How might this have influenced the conduct and analysis of this study? Has the relationship between researcher and participants been adequately considered? It would be helpful to see the researchers critically examine their own role, potential bias and influence during (a) formulation of the research questions (b) data collection, including sample recruitment and choice of location.

All authors of the study formulated the questions through a reflective, iterative and dialogic process. Respondents were given the lead in ‘setting the pace’ of the interview either over WhatsApp or face-to-face. Data collectors deliberately adopted a ‘back seat’ approach for participants to feel that they were exercising a measure of control over the interview process. During the data collection and recruitment, data collectors kept memos for each interview containing reflections, feelings towards the participant and potential biases introduced. A focus group with data collectors and the authors was conducted to understand how the data collectors’ interactions with participants might be influenced by their own professional background, experiences, and prior assumptions. Given the length of the current paper, we did not include these details in the manuscript, but would certainly do so if the editor will allow it.

-Are there any limitations in using different methods of data collection? Using WhatsApp is convenient but it may limit the depth of responses due to it being a burden to write out a long response.

There were no limitations in using the different data methods of data collection. Studies have shown that compared to other online modalities, adolescents prefer sharing information and communicating via instant messaging (IM) (Lee, 2007; Shapka, Domene, Khan, & Yang, 2016). Moreover, the results of this study concur with the results found in the study by Shapka et al (2016). Interviews conducted online produced fewer words and took longer to complete, however, there were no mean differences in the number and kind of themes that emerged or in the depth to which the themes were discussed. The findings suggest that despite taking longer and producing fewer words, data quality is unaffected by the mode of data collection

-Some more time spent outlining the importance of these findings is warranted. What new knowledge has been created? The introduction lines 58-61 outline some findings from research among school attending youth – these appear to be very similar to this study’s current findings/conclusions. What are the differences? This could be clearer in the discussion section.

Thank you for the comment. we have improved the discussion section by clearly stating from the outset the unique contributions of this study pertaining to OSY. The first paragraph of the discussion reads as:

The findings of this study have provided valuable insight into the profile of OSY at risk for alcohol and tobacco use. Overall, the social environment, which included friends and family, played a facilitating or inhibiting role in OSY’s alcohol and tobacco use. While similarities in social network determinants exist with in-school learners, this study additionally found that facilitators of alcohol and tobacco use include having more access to financial resources from peers and OSY having more OSY friends who may use alcohol and tobacco. OSY’s experience of the lack of parent support, coping with the loss of a parent and the larger role that extended family may play were also facilitators of alcohol and tobacco among OSY.

4. As above - despite there being no comparison with school attendees in this study, how might these results differ compared to previous research?

The findings of this research provide a better understanding of the composition of the OSY social network as well as the unique social network facilitators and inhibitors of alcohol and tobacco use among OSY. We have revised the discussion section to highlight how the findings of this study corroborate previous studies among in-school and OSY, as well as offer additional insights into the social network determinants of alcohol and tobacco use among OSY.

5.This study focuses on the social networks of OSY, but the wider determinants of smoking/alcohol need to be considered in the discussion. E.g. The availability of cigarettes/alcohol, any wider socio-economically patterns in usage, government policies influencing price/availability.

Thank you for the comment. We have revised the results and discussion section to include wider determinants of alcohol and tobacco use

The following additions were made:

Line 234: Despite national policy prohibiting the sale of cigarettes to minors, OSY were able to purchase cigarettes from the shop on their rout to the transport routes.

Line 278-280: Given national regulations prohibiting the sale of alcohol and tobacco to minors, older OSY may have also purchased the alcohol and tobacco on behalf of the group. 

Line 453-455: Enforcement of national policy prohibiting the sale of alcohol and tobacco to minors or raising the minimum age of tobacco and alcohol purchases may be beneficial.

Line 483: Studies have shown that socioeconomic disadvantaged adolescents are more likely to take up tobacco and alcohol use 

Line 486: South African policies exempting students who do not have the affordability from paying tuition fees should be enacted.

6.More discussion is warranted on contribution the study makes to existing knowledge or understanding. Use in intervention development is mentioned but what exactly would this look like? What targets have been created for intervention development? Why would an OSY tailored intervention be more beneficial than a more generic intervention targeting all youth (given that the social-network determinants appear to be similar between the two)

We thank the reviewer for the comments and used this suggestion to strengthen the paper. We included a “conclusions and implications” section highlighting the unique contributions of this study as well as possible targets for intervention. Attempts should be made to register all those who drop out of school to allow for tracking of school dropouts for intervention. Given that OSY do not have the protective factor of school-based interventions, interventions appropriate and accessible through the community is warranted. Furthermore, peer led programmes which have been shown to be successful among in-school learners could include a peer led OSY component in that encourage the diffusion of non-smoking and non-drinking norms in the community 

Reviewer #2: In this manuscript, the authors explored through qualitative research social network determinants of alcohol and tobacco use among out of school youth (OSY) in South Africa. They conducted 41 interviews with OSY in a South African urban area and their findings suggest that determinants of alcohol and tobacco use include drinking friends, family members close in age to OSY that drank and smoked and household issues. On the other hand, they found that religiosity, parental control over alcohol and tobacco, romantic partners, and non-smoking or non-drinking friends alleviated alcohol and tobacco use.

The topic is very interesting, but the manuscript needs more attention to the use of English language. Additionally, it could be improved in the following respects:

We thank the reviewer for the compliment and have used the comments to improve the manuscript

1)In the abstract the authors mention that ‘an important determinant of alcohol, and tobacco use is the adolescent's social network, which has globally not been explored among out of school youth (OSY). However, the paper only includes participants from South Africa.

We recognise that the sentence creates confusion and we therefore removed the word “globally”. 

2) It is also good practice for qualitative research to state the analytic method in the abstract.

The data was analysed using thematic content analysis, which has been added to the abstract

3)The last sentence of the abstract is very general. Authors only look at OSY, so their suggestions for interventions should be relevant to this group of people.

Thank you for the comment. We have improved the abstract by stating the following: understanding how the social network of OSY plays a role in alcohol and tobacco use is useful for gaining an insight into the profile of OSY at risk for alcohol and tobacco use. Registration of OSY youth and community-based peer led programmes that include influential OSY family and friends could be beneficial.

4) Authors should also give a better definition of adolescent and out of school youth regarding the age range. They provide information for different age groups in the Introduction and then they state that their sample included participants age between 13-20 years old. But there is no justification why they chose this age range.

Although the WHO defines adolescents as those between ages 10-19 years, this study has focused on adolescents between 13-20 years. A slightly older sample of adolescents was targeted because studies have shown that dropout tends to increase from the age of 15 (grade 9). Moreover, alcohol and tobacco use is usually initiated between 12-14 years and is highly prevalent in this age group. To our knowledge, no qualitative study has been conducted, specifically focusing on 13-20-year-old dropouts in South Africa. We added this detail to the introduction.

5) Authors present % of tobacco and alcohol use among school going learners and OSY. It would be clearer if they present these figures in the same paragraph.

We agree with your suggestion and moved the prevalence rates to reflect in the same paragraph.

6) There is no clear explanation why authors are focusing on alcohol and tobacco use and not other addictive behaviours?

National studies show that alcohol and tobacco are the most prevalent among adolescents compared to other addictive behaviours such as illegal and other drug use. We added this statement in the introduction and hope this provides clarity. While we acknowledge that other addictive behaviours warrant investigation, a separate study focusing would be beneficial as it may be burdensome on participants to cover all addictive behaviours in a single study. Moreover, eliciting specific information on alcohol and tobacco would be more meaningful for the development of specific interventions. 

7) It is not clear why authors needed 52 participants for their study and if they recruited more than 52. Additionally, why did authors initial recruit 4 smokers and 4 non-smokers and not 2 smokers, 2 non-smokers, 2 drinkers and 2 non-drinkers?

In order to complete the full respondent driven sampling process of two waves of recruitment with 8 seeds, it was proposed that up to 52 participants can be recruited. This has been made clearer in the manuscript. As indicated under the data collection section, 41 participants were sufficient as no new views were emanating from the interviews since the last three respondents. 

The initial focus of the paper was tobacco use among OSY and their social network. As the paper evolved, we focused on both alcohol and tobacco use. Despite targeting smokers and non-smokers as the initial seeds, we still acquired a sample that contains almost an even number of alcohol and tobacco users while minimising potential bias introduced, due to the nature of the respondent driven sampling strategy. 

8) How many interviews conducted in person and how many through whatsapp? The authors could perhaps also reflect on whether any differences in quality were observed across those interviewed via the different modalities.

12 respondents conducted the interview over a cellular phone application WhatsApp, and 29 respondents conducted the interview face-to-face. From our preliminary unpublished analysis, the content from both Whatsapp and face-to-face interviews were relevant. Studies have shown that compared to other online modalities, adolescents prefer sharing information and communicating via instant messaging (IM) (Lee, 2007; Shapka, Domene, Khan, & Yang, 2016). Moreover, the results of this study concur with the results found in the study by Shapka et al (2016). Interviews conducted online produced fewer words and took longer to complete, however, there were no mean differences in the number and kind of themes that emerged or in the depth to which the themes were discussed. The findings suggest that despite taking longer and producing fewer words, data quality is unaffected by the mode of data collection

9) Data collection tools would be clearer if they include examples of questions and probs (e.g. including the topic guide as a supplementary file). Did the authors assess participants’ alcohol and tobacco use and how?

Participants were asked to discuss their own alcohol and tobacco use and alcohol and tobacco use of their social network. These details have been added to the manuscript and we have included the full interview guide as a supplementary file.

10) The authors need to specify who gave them ethical approval.

Ethical approval was obtained from the Human Sciences Research Council. This detail has been added to the manuscript.

11) Results section: How many participants use both alcohol and tobacco?

We have included this in the table of sample characteristics

12) It is not clear if the friends of participants are also OSY or not.

We found that in general the participants had friends who were both in-school, and out of school

13) Results section could benefit with more examples, interview quotations to support the results. Especially results about parents the example provided did not support the findings. The authors also mention that ‘Often respondents hid their alcohol and tobacco use from their family or use little to no tobacco or alcohol in the presence of their older family members out of respect or fear’, but the example provided did not support such findings. It also appears that the analysis is a bit shallow. For example, the theme ‘Friends’ is very broad, and the presented quotation suggests that participants were bored and drinking/smoking because there isn’t much else to do. They also mentioned having resources/money to buy alcohol/tobacco, which is more of an ‘opportunity’ rather than ‘social’ variable that influences use. Again, the theme ‘Parent’ doesn’t quite seem to capture what’s expressed by participants (e.g. drinking due to lack of parental support/grief.

We thank the reviewers for their comment and have revised the results section. We have included subthemes under the broad themes “friends” and “family” to closely reflect and organise the data better. 

The subthemes for friends were initiation of alcohol and tobacco, financial resources, leisure boredom and resistance to tobacco and alcohol. Family subthemes were those of similar age, those elderly, lack of parental support and household norms. The results have been restructured to reflect these themes and sub-themes. More example quotations under each theme were added followed by a revised and in-depth analysis of the theme.

We provide additional quotes and context to smoking/drinking friends not playing a role in the initiation or use among the non-users. We further analysed non-users and it was found that they would engage in other leisure activities such as going to the mall, sports and attending church. They also claim that their friends did not excessively use alcohol and tobacco

We trust that the results section reads better.

14) Some discussion points are not supported by the findings i.e. ‘Among OSY who were current tobacco and alcohol users, OSY tended to spend more time with other OSY friends’; ‘ These findings may explain why a different sample of females, who dropped out of school due to financial difficulties in South Africa were smoking cigarettes, as seen in a previous study by Desai and colleagues (13)’. In the second example it is not also clear where different refers to?

Now that we have introduced the subtheme leisure boredom and included more quotation examples, this discussion point is supported. In the second example, the word “different” refers to the previous study’s sample not being the same sample as the current study. We rephrased this sentence to make it clear. We trust that the paragraph reads better with these changes. 

15) Authors should avoid use language such as substance users when they are only referring to alcohol and tobacco use

We have removed instances where we use this language in the methods, results and discussion section

16) Discussion about romantic partners is not clear. For example, the point that ‘compared to friends, OSY in this sample would smoke and drink less with their romantic partners, possibly due to spending less time with romantic partners in the context of smoking and drinking’ is not clear how is supported by the results.

Thank you for the comment. The way the results section has been reorganised, we have combined the discussion on romantic relationships with friends. Romantic partners are not the main finding of this study and therefore a full discussion on this point is not warranted. Further exploration of romantic relationships is needed in future studies. We trust that this presentation provides more clarity.

17) It would be useful to discuss implications for policy and avenues for future research.

We have included a conclusions and implications paragraph highlighting the recommendations for policy and programmes.

---

## [Decision Letter · Decision Letter 1]

26 Aug 2020

PONE-D-20-17565R1

Social network determinants of alcohol and tobacco use: a qualitative study among out of school youth in South Africa

PLOS ONE

Dear Dr. Desai,

Thank you for submitting your manuscript to PLOS ONE. After careful consideration, we feel that it has merit but does not fully meet PLOS ONE’s publication criteria as it currently stands. Therefore, we invite you to submit a revised version of the manuscript that addresses the points raised during the review process.

As you can see, Reviewer 2 has a number of additional suggestions for how to improve presentation and interpretation of your results (see below). In particular, please pay attention to 1) reporting transparently the change in focus of the study 2) ensure that presented themes and subthemes are supported by appropriate quotes and 3) discuss study limitations in greater detail.

We look forward to receiving your revised manuscript.

Kind regards,

Lion Shahab, MA MSc MSc PhD CPsychol

Academic Editor

PLOS ONE

Reviewers' comments:

Reviewer's Responses to Questions

**Comments to the Author**

1. If the authors have adequately addressed your comments raised in a previous round of review and you feel that this manuscript is now acceptable for publication, you may indicate that here to bypass the “Comments to the Author” section, enter your conflict of interest statement in the “Confidential to Editor” section, and submit your "Accept" recommendation.

Reviewer #1: All comments have been addressed

Reviewer #2: (No Response)

2. Is the manuscript technically sound, and do the data support the conclusions?

Reviewer #1: Yes

Reviewer #2: Partly

3. Has the statistical analysis been performed appropriately and rigorously? 

Reviewer #1: N/A

Reviewer #2: N/A

4. Have the authors made all data underlying the findings in their manuscript fully available?

Reviewer #1: Yes

Reviewer #2: Yes

5. Is the manuscript presented in an intelligible fashion and written in standard English?

Reviewer #1: Yes

Reviewer #2: Yes

6. Review Comments to the Author

Reviewer #1: The authors have responded well to the reviewer comments and made appropriate changes. If the editors will allow it, some more space to allow the authors to provide their justification for the whatsapp interview technique will strengthen the methods section.

Reviewer #2: The authors have been mostly responsive to the issues raised during the previous review. The manuscript has been improved, but a few comments remain:

1) Introduction, lines (44-45) ‘Alcohol and tobacco use, like in many other countries is prevalent among adolescents’. Not clear if authors are referring to LMICs or South Africa.

2) Authors provide a definition of adolescents, which defines adolescents as those aged 10-19 years old. Not clear why then they include participants 20 years old?

3) Authors state that they added this statement in the introduction ‘National studies show that alcohol and tobacco are the most prevalent among adolescents compared to other addictive behaviours such as illegal and other drug use’. But I can’t find it in the Introduction. Could they please indicate the lines?

4) In the methods section authors state (lines 148-151) ‘Initially, eight seeds were obtained, and they were required to identify up to two other OSY. These respondents recruited by the seeds formed the “first wave” of sampling and were themselves asked to identify and refer a further two more school dropouts (Fig 1). Up to two waves of recruitment were conducted. participants describe the method of recruitment’. Based on this description, the target should have been 56 participants and not 52 as stated in line 170.

5) Authors should also report the initial focus of the paper as mentioned in their response, in order to justify why they targeted only smokers and non-smokers.

6) Regarding my comment on ‘How many interviews conducted in person and how many through whatsapp? The authors could perhaps also reflect on whether any differences in quality were observed across those interviewed via the different modalities’. The authors provided the requested numbers, but they did not include any reflection in the manuscript.

7) Table 1 does not read well. % and number of whole sample should be closer together. Provide total numbers of males and females in the first line.

8) Authors have changed the Results section and they provided more examples to support their findings. They have provided new themes and sub-themes, but a few issues remain.

9) For example, the sub-theme ‘Initiation of alcohol and tobacco’ is only included in the friend section, however examples provided in sub-theme ‘family members of similar age’ of family section also supports the sub-theme ‘initiation of alcohol and tobacco’.

10) In line 352, authors state ‘Those respondents (n=8) who reported not using alcohol and/or tobacco still had smoking and/or drinking friends’. Not clear which group of participants are they referring to based on Table 1. Maybe the authors should state ‘Some of the respondents (n=8)…’

11) In the sub-theme ‘lack of parental support’ authors mention ‘Some respondents attributed their alcohol and tobacco use to not getting along with parents, feelings of neglect, grief and coping with household issues. Those neglected by their parents attributed their alcohol and tobacco use to the stress of contributing towards the household income’. However, the examples provided (lines 492-497) does not support such findings.

12) Sub-theme ‘resistance to alcohol and tobacco’. The quotations provided to support this sub-theme does not suggest resistance. Authors should consider a different title for this sub-theme.

13) Discussion about attributing alcohol and tobacco use to coping with household issues such as not being able to get along with parents, the stress of contributing towards the household income, feelings of neglect by parents and lack of parental support are not supported by results section.

14) Limitations of the study should discuss the different methods of data collection.

7. PLOS authors have the option to publish the peer review history of their article (what does this mean?). If published, this will include your full peer review and any attached files.

Reviewer #1: **Yes: **Dr Loren Kock

Reviewer #2: No

---

## [Author Response · Author response to Decision Letter 1]

29 Sep 2020

Reviewer #1: The authors have responded well to the reviewer comments and made appropriate changes. If the editors will allow it, some more space to allow the authors to provide their justification for the WhatsApp interview technique will strengthen the methods section.

We thank the reviewer for the constructive feedback. We included a justification for the WhatsApp interview technique in the methods section under the sub-heading “data collection” on page 8. 

Reviewer #2: The authors have been mostly responsive to the issues raised during the previous review. The manuscript has been improved, but a few comments remain:

We thank the reviewer for the comments and used them to further improve the quality of the paper. 

1) Introduction, lines (44-45) ‘Alcohol and tobacco use, like in many other countries is prevalent among adolescents’. Not clear if authors are referring to LMICs or South Africa. 

We are refereeing to LMIC’s. This clarification has been made in line 44

2) Authors provide a definition of adolescents, which defines adolescents as those aged 10-19 years old. Not clear why then they include participants 20 years old?

In the South African education system, repetition rates are known to be high from Grade 9 up to Grade 11 resulting in students’ older students attending high school. We therefore feel it is appropriate to include participants 20 years old. We have included this detail in line 105.

3) Authors state that they added this statement in the introduction ‘National studies show that alcohol and tobacco are the most prevalent among adolescents compared to other addictive behaviours such as illegal and other drug use’. But I can’t find it in the Introduction. Could they please indicate the lines?

We thank the reviewer for pointing this out and added this statement to lines 107-108. 

4) In the methods section authors state (lines 148-151) ‘Initially, eight seeds were obtained, and they were required to identify up to two other OSY. These respondents recruited by the seeds formed the “first wave” of sampling and were themselves asked to identify and refer a further two more school dropouts (Fig 1). Up to two waves of recruitment were conducted. participants describe the method of recruitment’. Based on this description, the target should have been 56 participants and not 52 as stated in line 170.

We thank the reviewer for pointing this out and corrected this to 56

5) Authors should also report the initial focus of the paper as mentioned in their response, in order to justify why they targeted only smokers and non-smokers.

We have included this statement in lines 146-151.

6) Regarding my comment on ‘How many interviews conducted in person and how many through whatsapp? The authors could perhaps also reflect on whether any differences in quality were observed across those interviewed via the different modalities’. The authors provided the requested numbers, but they did not include any reflection in the manuscript.

We thank the reviewer for the constructive feedback. Despite taking longer and producing fewer words in the online condition, data quality was unaffected by the mode of data collection (online versus face-to-face) with no differences in the number, depth and type of themes discussed

In lines 158-173, we included a justification for the WhatsApp interview technique as well as a reflection on the quality of the different interviewing methods. 

7) Table 1 does not read well. % and number of whole sample should be closer together. Provide total numbers of males and females in the first line.

We have made the recommended adjustments and trust that the table reads better

8) Authors have changed the Results section and they provided more examples to support their findings. They have provided new themes and sub-themes, but a few issues remain.

9) For example, the sub-theme ‘Initiation of alcohol and tobacco’ is only included in the friend section, however examples provided in sub-theme ‘family members of similar age’ of family section also supports the sub-theme ‘initiation of alcohol and tobacco’.

We acknowledge that sections of data may be included in multiple themes with some overlap between themes. However, the researchers considered how each theme fit into the overall story about the entire data set. Although there were some instances on initiation with family, we feel that this quote also supports the theme of “family members of similar age”. We have included this statement in lines 543. We trust that this decision still allows for the paper to read well.

10) In line 352, authors state ‘Those respondents (n=8) who reported not using alcohol and/or tobacco still had smoking and/or drinking friends’. Not clear which group of participants are they referring to based on Table 1. Maybe the authors should state ‘Some of the respondents (n=8)…’

We have made the adjustment in the manuscript.

11) In the sub-theme ‘lack of parental support’ authors mention ‘Some respondents attributed their alcohol and tobacco use to not getting along with parents, feelings of neglect, grief and coping with household issues. Those neglected by their parents attributed their alcohol and tobacco use to the stress of contributing towards the household income’. However, the examples provided (lines 492-497) does not support such findings.

We have included an additional quote where respondents attributed their alcohol and tobacco use to not getting along with parents and coping with household issues. We feel that the other quotations capture parental neglect, grief and the stress of contributing towards the household income. We trust that the presentation of quotes support the findings.

12) Sub-theme ‘resistance to alcohol and tobacco’. The quotations provided to support this sub-theme does not suggest resistance. Authors should consider a different title for this sub-theme.

We have changed this subtheme to reluctance to alcohol and tobacco

13) Discussion about attributing alcohol and tobacco use to coping with household issues such as not being able to get along with parents, the stress of contributing towards the household income, feelings of neglect by parents and lack of parental support are not supported by results section.

We hope that the revised presentation of quotes for this theme on page 20 now support the discussion points.

14) Limitations of the study should discuss the different methods of data collection.

We thank the reviewer for this suggestion. The different interviewer methods may have influenced participant responses. However similar to the study by Shapka et al (2016), the findings of this study suggest that data quality is unaffected by the mode of data collection. We have added this statement in lines 541.

---

## [Editor Report · Decision Letter 2]

1 Oct 2020

Social network determinants of alcohol and tobacco use: a qualitative study among out of school youth in South Africa

PONE-D-20-17565R2

Dear Dr. Desai,

We’re pleased to inform you that your manuscript has been judged scientifically suitable for publication and will be formally accepted for publication once it meets all outstanding technical requirements.

Kind regards,

Lion Shahab, MA MSc MSc PhD CPsychol

Academic Editor

PLOS ONE
---

## [Editor Report · Acceptance letter]

9 Oct 2020

PONE-D-20-17565R2 

Social network determinants of alcohol and tobacco use: a qualitative study among out of school youth in South Africa 

Dear Dr. Desai:

I'm pleased to inform you that your manuscript has been deemed suitable for publication in PLOS ONE. Congratulations! Your manuscript is now with our production department. 

Kind regards, 

on behalf of

Dr. Lion Shahab 

Academic Editor

PLOS ONE